# Effects of Extending Milk Replacer Feeding during the Fattening Period on the Behaviour and Welfare of Lambs: A Preliminary Study

**DOI:** 10.3390/ani13010085

**Published:** 2022-12-26

**Authors:** Ana González-Martínez, Andrés Luis Martínez Marín, Rubén Lucena, Miriam González-Serrano, Miguel Ángel de la Fuente, Pilar Gómez-Cortés, Evangelina Rodero

**Affiliations:** 1Department of Animal Production, Faculty of Veterinary Sciences, University of Cordoba, 14071 Córdoba, Spain; 2Instituto de Investigación en Ciencias de la Alimentación (CIAL, CSIC-UAM), Nicolás Cabrera, 9, 28049 Madrid, Spain

**Keywords:** ethogram, behaviour, welfare, lambs, fattening, milk replacer

## Abstract

**Simple Summary:**

Artificial milk feeding in fattening lambs would promote the functionality of the reticular groove reflex in order to avoid the rumen fermentation of selected feed supplements included in the milk replacer. However, there is a lack of information of the behavioural and welfare effects of such practice. Therefore, we studied the effects of extending or not milk replacer feeding during fattening with a high concentrate diet on the behaviour of 16 non castrated male lambs of the Manchega breed (eight lambs were in the group that were fed daily a bottle of milk, and the other eight were in the fully weaned group) by means of a video recording system. Both groups consumed the same solid diet (concentrate plus cereal straw) and were kept under the same conditions. Weaned lambs showed longer resting episodes and a higher frequency of self-grooming, whereas unweaned lambs showed a higher frequency of interaction behaviour. From the interpretation of such behaviours in the conditions of the present study, we concluded that the welfare status was almost similar in both groups.

**Abstract:**

There is a lack of information on the behavioural and welfare effects of sustaining artificial milk feeding in fattening lambs. Therefore, the present work aimed to study the effects of prolonged artificial milk feeding during fattening with a high concentrate diet on the behaviour of lambs. The behaviour of 16 non castrated male lambs of the Manchega sheep breed (eight lambs were in the group that were fed daily a bottle of milk, and the other eight were in the weaned group) was recorded with four fixed cameras just before bottle feeding (~8:30 a.m.) of the unweaned group till four hours later, every day for 7 weeks. The solid diet (pelleted concentrate plus cereal straw) and housing conditions were the same in both groups. Solid feeds were offered ad libitum. There were no differences between groups in time spent eating nor in drinking, playing, scratching and oral activity behaviours (*p* > 0.05), but resting episodes were longer in weaned lambs (*p* < 0.05). Weaned lambs presented a higher frequency of self-grooming behaviour (*p* < 0.05), while the unweaned group performed a higher frequency of interaction behaviour (*p* < 0.05). In conclusion, the behaviours of lambs that were fed daily a bottle of milk during the fattening period did not substantially differ from the weaned individuals.

## 1. Introduction

The behaviour of lambs can be affected by many endogenous and exogenous factors, including the type of feeding and management and housing conditions [1]. In sheep milk production systems, the rearing of lambs with milk replacer instead of maternal milk is a common practice in order to maximize saleable milk [2,3]. Weaning is one of the most critical moments for lambs, since it involves social, nutritional and environmental changes [4]. Natural weaning is a slow process, which takes place between 100 and 150 days of life, in which the animal gradually transitions from milk feeding to solid feeding. The artificial weaning means forcibly ending the sucking of maternal milk by the offspring, which may affect animal welfare. In both weaning types, when lambs are regrouped with other lambs of similar age, either on pasture or in feedlots, they develop a strong attachment to their peers, which influences their behaviour [5,6]. Keeping the lambs in groups after weaning helps in their socialization and can reduce abnormal behaviour [7]. Behavioural effects of rearing lambs with either their dams or milk replacer plus solid feed until slaughter at 45 days of age (no weaning) has been previously reported [8]. Herath et al. [9] studied the effects of feeding milk replacer along with pelleted concentrates until slaughter at 57 days of age, with or without weaning at 42 days of age, on growth performance and body composition. However, there is a lack of information on the behavioural effects of prolonged artificial milk feeding during fattening with a high concentrate diet. One reason to sustain this artificial milk feeding is to keep functional the reticular groove reflex to avoid rumen fermentation of selected feed supplements included in the milk replacer [10,11]. In this line, it has been shown that supplementing algae meal within the milk replacer has remarkable effects on lamb meat quality (i.e., modifying meat fatty acid profile and volatile compounds) [12,13], but the behavioural effects, if any, have not been described yet. Moreover, the observation of lamb behaviour can provide information on their preferences, requirements and internal states, which should be taken into account to improve farm animal welfare [1].

The aim of the present work was to study the effects of extending milk replacer feeding along with solid feed during the fattening period on the behaviour and welfare of lambs. 

## 2. Materials and Methods

### 2.1. Animals, Housing and General Management

#### 2.1.1. Initial Handling of Lambs

A total of 16 non castrated male lambs of the Manchega breed were used. Typically, lambs from the Manchega sheep breed, one of the main Spanish autochthonous sheep breeds, are removed from their mothers after colostration until 3–5 days of age, being reared from then on with milk replacer. Experimental lambs were purchased in a commercial farm and transferred to the same conditioned room (equipped with heaters, forced ventilation and natural and artificial lighting) in the premises of the Animal Production building at the University of Córdoba (Spain) just after colostrum feeding. The lambs were bottle-fed milk replacer ad libitum, a couple of times per day, up to 28 days of age. In order to stimulate the rumen development, from 15 days of age onwards, a concentrate, based on cereals and soybean meal and formulated for growing lambs (16% crude protein, 11 MJ/kg as fed), was offered ad libitum as well. At 28 days of age, the daily volume of milk replacer was reduced to 500 mL, in a single shot. The aim was to foster solid feed consumption. After 7 days, the daily volume of milk replacer was further reduced to 250 mL. Milk replacer (Vigolait Active, Iniciativas Alimentarias S.A., Ciudad Real, Spain) was prepared as the package leaflet recommended. 

#### 2.1.2. Management of Animals throughout the Experiment

At 42 days of age, lambs were weighed, painted with large dorsal numbers, and assigned to one of eight pens, in pairs of similar body weight. The pens were 1.40 m^2^ raised, slatted floor cages with individual troughs for feed and water. Afterward, all the pens were weighed. Then they were blocked by pairs in four groups attending to the recorded weights. Within each pair, the pens were randomly allocated to one of two groups (four pens per group and a total of eight animals per group), namely weaned and unweaned. Hence, all the weights were represented in both treatments. In contrast, feeding of the individuals of the unweaned group was complemented with 250 mL daily of milk replacer in a single shot until slaughter. Final experimental body weight was fixed at ~25 kg as the average and the experimental period lasted seven weeks (between 42 and 91 days of age). During the experimental period, body weight per pen was recorded weekly and feed intake per pen was recorded daily. Then, the average daily gain (g/day), average feed intake (g/day) and feed conversion ratio (feed to gain g/g) were calculated for each pen. The animals were clinically examined at the beginning and throughout the experiment to ensure that they were healthy, there was no loss due to death or disease.

### 2.2. Data Collection

A set of four fixed video cameras (AXIS M2014-E), controlled by a computer running Noldus Media Recorder XT^®^ software (Noldus, Wageningen, The Netherlands) were placed in front of two pens each at a height of approximately 2.10 m. In order to prevent other stimuli interfering in the test, the time for the observations was determined by the interval between the activities of replacing the concentrate for the animals and those of cleaning the facilities. Thus, the behaviour of all lambs was recorded just after bottle feeding (~8:30 a.m.) till four hours later. All the video recordings were analysed by means of the Noldus© Observer XT 12 software (Noldus, Wageningen, The Netherlands) by the same trained observer. The behaviours studied were eating (eating either pelleted concentrate or cereal straw from the feed through), resting (lamb lying down), drinking (lamb swallowing water from the water through), self-grooming (lamb licking itself with the tongue or rubbing its coat with the teeth), interaction (lamb sniffing or grooming partners), playing (lamb butting, nudging, kicking or tupping partners), scratching (lamb rubbing itself with its hind hooves or against pen structures) and oral activity (lamb sucking the pen structures). Length or counts of the behaviours were transferred to an Excel spreadsheet (Microsoft Inc., Redmond, WA, USA).

### 2.3. Statistical Analysis

SAS UE 3.8 (SAS Institute Inc., Cary, NC, USA) was used to perform the statistical analyses. All behaviours were considered quantitative variables, i.e., we measured the average time (s) that the lambs spent eating and resting, and the frequencies (%) of drinking, grooming, interaction, playing, scratching and oral activities. A repeated measurements analysis was conducted with the MIXED procedure [14]. The statistical model included the fixed effects of group (weaned and unweaned), fattening phase (weeks 1 to 7) and their interaction; the repeated effect was time; the subject of the repeated measurements was the animal nested within group and pen. When the fixed effects were significant, differences between least squares means were assessed by paired *t*-test.

## 3. Results and Discussion

Average feed intake, daily body weight gain and feed conversion ratio did not differ between experimental groups [15]. The values were 806 ± 49.7 g, 321 ± 34.0 g and 2.5 ± 0.24 g/g, for average feed intake, daily body weight gain and feed conversion ratio respectively. Those values are in agreement with previous studies on the same breed, age range and intensive feeding conditions, except for milk replacer feeding [16]. Experimental animals were homogeneous in initial age, weight (11.3 ± 1.69 and 11.2 ± 1.43 kg body weight, respectively; *p* = 0.91) and sex, as well as they were maintained under the same housing and management conditions. Thus, any differences in the analysed behaviours with the help of Observer XT^®^ 12 software should be only ascribed to the effects of group (weaned vs. unweaned), time on fattening and their interaction. This software has proved to be useful in behavioural research with sheep, such as studies focused on the self-organization of flocks at pasture and classification of behaviours of grazing, ruminating and non-eating and other biologically relevant behaviours [17,18,19].

Table 1 shows the effects of group and time (week) on fattening and their interaction on frequency of drinking behaviour and average duration of each episode elicited in eating and resting behaviours. Average duration of eating and resting episodes was 116 ± 34.5 and 470 ± 165.9 s, respectively. There were no differences between groups (*p =* 0.67) in time spent eating in each episode. The duration of eating episodes showed an increasing linear trend over the experimental period in both groups, as might be expected from the raising feed intake of the lambs in order to meet their nutritional requirements for growth [20]. The increase in eating duration was more pronounced during the first two weeks especially in the weaned group, which would suggest a quicker adaptation of this group to the only-solid feeding to satisfy their nutritional demands for body growth [9]. Resting episodes were longer in weaned lambs (526 ± 167.2 vs. 422 ± 149.8 s, *p* < 0.05). Resting is a high priority and an inelastic behavioural need in ruminants and longer resting times in the absence of illness can be associated with a positive state of welfare [21]. Moreover, a longer resting time in unweaned lambs has been attributed to the absence of distress that the separation from the mother causes [22]. Nevertheless, rather than ascribing this behaviour to an inferior welfare in the unweaned group, the present results should be ascribed to the loss of the human–lamb bond after weaning and the subsequent lack of expectation in milk feeding of the weaned group [23]. One should bear in mind that the recordings began just at the time of preparation routines of the milk replacer, thus associated sounds and smells would anticipate milk feeding in the unweaned lambs thereby exciting them. In this manner, milk feeding acted as a reward that elicited an anticipatory behaviour [24]. The frequency of drinking episodes did not differ between groups (5.5 ± 4.58%; *p* = 0.15). Mohapatra et al. [22] found that weaned lambs spent more time drinking, probably due to water supplied by the consumption of milk in the unweaned lambs.

Average frequencies of self-grooming, interaction, playing, scratching and oral activity behaviours were 39 ± 14.3, 27 ± 12.1, 4.0 ± 3.3, 8.3 ± 4.76 and 17 ± 9.3%, respectively. There were no differences (*p >* 0.05) in the frequencies of playing, scratching and oral activity behaviours between groups (Table 2). On the contrary, Mohapatra et al. [22] observed that weaned lambs exhibited shorter playing time than unweaned lambs kept with their mothers from 17:00 p.m. to 9:00 a.m. Playing behaviour is considered as a yardstick to measure animal welfare [25], thus in the present study a diminished welfare as a result of ceasing milk feeding in weaned lambs did not occur. Moreover, the absence of differences in scratching and oral activity behaviours would indicate that the welfare status was similar between groups [26,27,28]. The significant differences of playing, scratching and oral activity behaviours over the experimental period suggests that they were more related to age-dependent factors than to the cessation of milk feeding [25,27]. 

Weaned lambs presented a higher frequency of self-grooming behaviour than unweaned lambs (44 ± 15.1% vs. 34 ± 11.6%) (Table 2). In calves, self-grooming is considered a comfort activity facilitated by a greater space availability in the stalls [29]. In the present study, the available space for the lambs within the pens was the same, thus the higher number of self-grooming episodes in the weaned group might be due to differences in spare time between groups [30], i.e., unweaned lambs spent more time paying attention to the supply of the milk replacer, which would be in concordance with the lower resting time in that group. In agreement with previous studies in goats and sheep, self-grooming behaviour decreased with age (*p* < 0.05) [31].

The unweaned group performed a higher frequency of interaction behaviour than the weaned one (30 ± 11.8% vs. 24 ± 11.7%,) (Table 2), which might be related to the need to compensate for the reduced suckling stimulus provided by the short time spent while bottle feeding. Affiliative behaviours included within the so-called “interaction” have been also related with the supply of straw [32] but both groups in the present study were offered straw ad libitum. The manifestation of affiliative relationships between lambs improves their welfare by reducing aggressive relationships [33]. In this regard, the performance of affiliative behaviours by the unweaned lambs would allow them to reduce the stress caused by the demand for the bottle. Affiliative behaviours can be used as an indicator of positive experiences in farm animals [34], but they are also a way of conflict resolution [35]. Therefore, the increase of interaction behaviour over time (*p* < 0.05) in the present study might be interpreted as a means used by the lambs to reduce the social tension derived by the diminishing available space as their body size increased. 

## 4. Conclusions

Most of the behaviours of lambs that were fed daily a bottle of milk during the fattening period under intensive feeding conditions were not substantially modified in comparison with weaned lambs. The results of the present study also suggest that, at least, a four-week period of adaptation to the experimental conditions is required to maintain the welfare status. New research increasing the number of animals as well as behavioural and performance parameters would be needed to shed more light on this issue.

## Figures and Tables

**Table 1 animals-13-00085-t001:** Average length of eating and resting episodes (in seconds) and frequency of drinking behaviour (in percentage) of lambs fed milk replacer (UNWEAN) or not (WEAN) during the fattening period.

Behaviour	Group (G)	Week (W)							SEM ^1^	*p*		
		1	2	3	4	5	6	7		G	W	G × W
Eating	WEAN	81 ^C^	125 ^AB^	125 ^AB^	136 ^A^	117 ^AB^	112 ^B^	135 ^A^	3.3	0.65	<0.001	0.26
	UNWEAN	78 ^B^	96 ^B^	121 ^A^	132 ^A^	107 ^AB^	122 ^A^	143 ^A^				
Resting	WEAN	631 ^AB^	455 ^BCDa^	350 ^D^	672 ^Aa^	500 ^C^	535 ^BC^	539 ^BC^	16.2	<0.05	<0.001	<0.05
	UNWEAN	577 ^A^	321 ^Bb^	243 ^B^	410 ^Bb^	398 ^B^	465 ^B^	538 ^A^				
Drinking	WEAN	7.5 ^AB^	5.8 ^AB^	7.1 ^AB^	10 ^A^	8.3 ^AB^	4.7 ^B^	6.9 ^AB^	0.05	0.15	< 0.01	0.75
	UNWEAN	3.0 ^C^	3.5 ^BC^	6.8 ^AB^	7.1 ^A^	7.2 ^A^	1.3 ^C^	3.4 ^BC^				

^ABC^ For each behaviour, within a group, least square means without a common superscript differ significantly (*p* < 0.05) between weeks. ^ab^ For each behaviour, within a week, least square means without a common superscript differ significantly (*p* < 0.05) between groups. ^1^ Standard error of the mean.

**Table 2 animals-13-00085-t002:** Frequency of behaviours elicited (in percentage) of lambs fed (unweaned, UNWEAN) or not (weaned, WEAN) with milk replacer during the seven weeks of fattening period time.

Behaviour	Group (G)	Week (W)							SEM ^1^	*p*		
		1	2	3	4	5	6	7		G	W	G × W
Self-grooming	WEAN	41 ^B^	51 ^ABa^	41 ^B^	55 ^A^	55 ^Aa^	40 ^Ba^	44 ^AB^	1.4	<0.05	<0.001	0.67
	UNWEAN	31 ^B^	37 ^ABb^	36 ^AB^	44 ^A^	35 ^ABb^	25 ^Bb^	31 ^B^				
Interaction	WEAN	19 ^BC^	18 ^C^	20 ^BC^	18 ^C^	22 ^BCb^	36 ^A^	28 ^AB^	1.2	<0.05	<0.001	0.80
	UNWEAN	23 ^C^	27 ^BC^	26 ^BC^	21 ^C^	35 ^Aa^	46 ^A^	33 ^B^				
Playing	WEAN	2.6 ^BC^	3.4 ^BC^	1.4 ^C^	2.9 ^BC^	3.3 ^BC^	4.8 ^B^	8.7 ^A^	0.35	0.79	<0.01	0.06
	UNWEAN	6.6 ^A^	3.7 ^AB^	1.5 ^B^	4.3 ^AB^	3.0 ^B^	4.7 ^AB^	4.5 ^AB^				
Scratching	WEAN	9.5 ^A^	7.4 ^AB^	9.7 ^A^	9.4 ^A^	7.2 ^AB^	4.7 ^B^	9.4 ^A^	0.46	0.86	<0.05	0.82
	UNWEAN	11 ^A^	9.4 ^AB^	7.6 ^AB^	9.8 ^AB^	6.4 ^B^	6.0 ^B^	8.6 ^AB^				
Oral activity	WEAN	24 ^A^	24 ^A^	27 ^A^	13 ^B^	12 ^B^	12 ^B^	11 ^B^	0.1	0.57	<0.001	0.23
	UNWEAN	22 ^A^	19 ^A^	20 ^A^	13 ^B^	10 ^B^	12 ^B^	16 ^AB^				

^ABC^ For each behaviour, within a group, least square means without a common superscript differ significantly (*p* < 0.05) between weeks. ^ab^ For each behaviour, within a week, least square means without a common superscript differ significantly (*p* < 0.05) between groups. ^1^ Standard error of the mean.

## Data Availability

The data presented in this study are available in the article.

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
