# Peer review of "Effects of Extending Milk Replacer Feeding during the Fattening Period on the Behaviour and Welfare of Lambs: A Preliminary Study"

_animals, 2022, doi:10.3390/ani13010085_

Round 1

Reviewer 1 Report

General Comments

In the manuscript entitled “Behavioural and welfare assessment of fattening lambs fed milk replacer”, González Martínez et al., studied the effects of extending or not milk replacer feeding during fattening on the behaviour and welfare status of lambs. The topic is of interest, as there is a lack of information about it, while the effect on growth performance and body composition has  been studied. In my opinion, these results, although interesting, are preliminary and do not have sufficient entity to be presented as a research article (3000 words at minimum). A larger number of animals should be studied, and additional information on laboratory parameters such as stress markers could also help to reach a clearer conclusion. The results presented here would fit better as a short paper, however, a major revision of the paper in its present form is needed in order to improve its quality and comprehension.

 Major concerns

-          The conclusion does not take into account some results. Authors conclude (lines 34-36) that the results support the welfare status of lambs that continue to receive milk replacer during the fattening period under intensive feeding conditions is not superior nor worse than that of weaned lambs. This is not completely true because although no differences between both groups were detected in several behaviours studied, significant differences were found in others (resting episodes, frequency of self-grooming behaviour and frequency of interaction behaviour). The authors attributed these differences to the fact that unweaned lambs spent more time paying attention to milk substitute supply (lines 185-186) and their need to compensate for the reduced lactation stimulus provided by the short time spent during bottle feeding (lines 190-192) and to reduce stress caused by bottle demand (lines 196-197). Could this be interpreted as in some extent the welfare of unweaned lambs is lower? This must be clarified in the article.

-          Were the two groups of lambs, weaned and unweaned, accommodated in the same room during the experimental period? This aspect is not clearly specified in the article, although it is stated that experimental lambs were transferred from a commercial farm to a conditioned room… (lines 74-77). If both groups were in the same room, is it possible that the sounds and smells associated with the preparation of milk replacer and the fact that weaned lambs do not receive the reward of milk while the other group does, somehow affect the behavior of weaned lambs?

-          Were the lambs clinically examined throughout the study? This is not indicated in the article, and the presence of illness could affect the behaviors studied.

-          In lines 127-128 authors state that both groups were homogeneous in initial age, weight, and sex. Does it mean that significant differences between the groups were not found in relation to initial weight? This must be clarified in the text as differences in initial weight could affect the duration of eating episodes.

-          In the Materials and Methods section it is stated  that during the experimental period, body weight per pen was recorded weekly and feed intake per pen was recorded daily, and then, average daily gain (g/day), average feed intake (g/day), and feed conversion ratio (feed to gain g/g) were calculated for each pen (lines 96-99). However these results are not shown in the article, authors only indicate, in a general way for all lambs in the study, that average feed intake, daily body weight gain and feed conversion ratio were 806 ± 125 49.7 g, 321 ± 34.0 g and 2.5 ± 0.24 g/g, respectively, which is in agreement with previous studies on the same breed, age range and intensive feeding conditions (lines 125-127). Were there differences between the two groups regarding these parameters? Were these results in agreement with those from other authors that studied the effects of feeding milk replacer along with concentrates, with or without weaning, on growth performance? The quality and interest of the article would improve with the inclusion of these data.

Minor concerns

- The meaning of sentence in lines 88-91 could be confusing to readers. What exactly does it mean that the pens were blocked in four groups by their average body weight? How many groups were studied? four, two? Were all animals in the same block assigned to the same group (weaned or unweaned)?

- The meaning of lowercase letters superscripts must be indicated in the Table 1.

- In lines 187-188, it is stated that, in agreement with previous studies, self-grooming behaviour was influenced by age. Could you explain in more detail what the effect of age is?

- Ethical approval.  The paper should contain details of approval by a properly constituted research ethics committee.

Author Response

RESPONSE

We would like to thank for your work and valuable comments that have substantially helped us to improve the former manuscript quality. All your comments have been considered and corrections have been made according to them. To make your work easier, we have highlighted the changes you and rest of reviewers requested by using track changes function.

We hope that you like the new version of the manuscript.

Thank you very much for your interest.

Kind Regards,

The authors

1/ General Comments

In the manuscript entitled “Behavioural and welfare assessment of fattening lambs fed milk replacer”, González Martínez et al., studied the effects of extending or not milk replacer feeding during fattening on the behaviour and welfare status of lambs. The topic is of interest, as there is a lack of information about it, while the effect on growth performance and body composition has  been studied. In my opinion, these results, although interesting, are preliminary and do not have sufficient entity to be presented as a research article (3000 words at minimum). A larger number of animals should be studied, and additional information on laboratory parameters such as stress markers could also help to reach a clearer conclusion. The results presented here would fit better as a short paper, however, a major revision of the paper in its present form is needed in order to improve its quality and comprehension.

Major concerns

2/ The conclusion does not take into account some results. Authors conclude (lines 34-36) that the results support the welfare status of lambs that continue to receive milk replacer during the fattening period under intensive feeding conditions is not superior nor worse than that of weaned lambs. This is not completely true because although no differences between both groups were detected in several behaviours studied, significant differences were found in others (resting episodes, frequency of self-grooming behaviour and frequency of interaction behaviour). The authors attributed these differences to the fact that unweaned lambs spent more time paying attention to milk substitute supply (lines 185-186) and their need to compensate for the reduced lactation stimulus provided by the short time spent during bottle feeding (lines 190-192) and to reduce stress caused by bottle demand (lines 196-197). Could this be interpreted as in some extent the welfare of unweaned lambs is lower? This must be clarified in the article.

Answer: We have rewritten the conclusions as following: “Most of the behaviors of lambs that were fed daily a bottle of milk during the fattening period under intensive feeding conditions are not substantially modified in comparison with weaned lambs. The results of the present study also suggest that, at least, four weeks period of adaptation to the experimental conditions would be required to maintain the welfare status. New research increasing the number of animals as well as behavioral and performance parameters would be needed to shed more light on this issue.”

3/ Were the two groups of lambs, weaned and unweaned, accommodated in the same room during the experimental period? This aspect is not clearly specified in the article, although it is stated that in experimental lambs were transferred from a commercial farm to a conditioned room… (lines 74-77). If both groups were in the same room, is it possible that the sounds and smells associated with the preparation of milk replacer and the fact that weaned lambs do not receive the reward of milk while the other group does, somehow affect the behavior of weaned lambs?

Answer: Yes, all animals were housed in the same room as clarified in the M&M section of the new version. As was discussed in the original version, there was no stimuli in the unweaned group due to the preparation of milk replacer. See lines 167-174 in the new version.

4/ Were the lambs clinically examined throughout the study? This is not indicated in the article, and the presence of illness could affect the behaviors studied.

Answer: We agree with you. A new sentence has been inserted in the new version of the manuscript to cover this query: “The animals were clinically examined at the beginning and throughout the experiment to ensure that they were healthy, there was no loss due to death or disease.”

5/ In lines 127-128 authors state that both groups were homogeneous in initial age, weight, and sex. Does it mean that significant differences between the groups were not found in relation to initial weight? This must be clarified in the text as differences in initial weight could affect the duration of eating episodes.

Answer: The initial weight did not differ between groups. This has been now included in lines 146-148 of the R&D section.

6/ In the Materials and Methods section it is stated  that during the experimental period, body weight per pen was recorded weekly and feed intake per pen was recorded daily, and then, average daily gain (g/day), average feed intake (g/day), and feed conversion ratio (feed to gain g/g) were calculated for each pen (lines 96-99). However these results are not shown in the article, authors only indicate, in a general way for all lambs in the study, that average feed intake, daily body weight gain and feed conversion ratio were 806 ± 125 49.7 g, 321 ± 34.0 g and 2.5 ± 0.24 g/g, respectively, which is in agreement with previous studies on the same breed, age range and intensive feeding conditions (lines 125-127). Were there differences between the two groups regarding these parameters? Were these results in agreement with those from other authors that studied the effects of feeding milk replacer along with concentrates, with or without weaning, on growth performance? The quality and interest of the article would improve with the inclusion of these data.

Answer: This query has been responded in the lines 141-144 at the beginning of the R&D section. New data can be found in those lines.

7/ The meaning of sentence in lines 88-91 could be confusing to readers. What exactly does it mean that the pens were blocked in four groups by their average body weight? How many groups were studied? four, two? Were all animals in the same block assigned to the same group (weaned or unweaned)?

Answer: The paragraph has been rewritten for improve the understanding as following: “Afterward, all the pens were weighed. Then they were blocked by pairs in four groups at-tending to the recorded weights. Within each pair, the pens were randomly allocated to one of two groups (four pens per group and a total of eight animals per group), namely weaned and unweaned. So, all the weights were represented in both treatments.”

8/ The meaning of lowercase letters superscripts must be indicated in the Table 1.

Answer: Correction done

9/ In lines 187-188, it is stated that, in agreement with previous studies, self-grooming behaviour was influenced by age. Could you explain in more detail what the effect of age is?

Answer: The sentence has been rewritten to clarify the point.

10/ Ethical approval.  The paper should contain details of approval by a properly constituted research ethics committee.

Answer: The information was included in the appropriate section of the original manuscript. We have now attached the Ethical approval with the revised version of the manuscript.

Reviewer 2 Report

line 77-78 - Were the animals of the experimental group fed twice a day throughout the experiment? If so, please provide justification for such action. According to the various available feeding schedules, this should be changed with age

line 103-104 - please justify the examination for 4 hours after feeding, why did the authors not decide on the scheme of 2 recordings?

line 107-111- Was excretion behavior analyzed? if so, please add it in the method

line 125-126 please specify average daily gain, average feed intake and feed conversion ratio in particular groups. Were leftovers of uneaten feed controlled?

line 135-136 in the description of table 1, please explain whether the time spent on individual activities in a given week is given as a daily or weekly result?

Author Response

RESPONSE

We would like to thank for your work and valuable comments that have substantially helped us to improve the former manuscript quality. All your comments have been considered and corrections have been made according to them. To make your work easier, we have highlighted the changes you and rest of reviewers requested by using track changes function.

We hope that you like the new version of the manuscript.

Thank you very much for your interest.

Kind Regards,

The authors

1/ line 77-78 - Were the animals of the experimental group fed twice a day throughout the experiment? If so, please provide justification for such action. According to the various available feeding schedules, this should be changed with age

Answer: No. From the beginning of the experimental period, milk replacer was fed once a day in the morning. Please check the subsection 2.1.2 in the M&M section.

2/ line 103-104 - please justify the examination for 4 hours after feeding, why did the authors not decide on the scheme of 2 recordings?

Answer: This was motivated to prevent other stimuli interfering in the experience. In this sense, we have added the following phrase in the text: “In order to prevent other stimuli interfering in the test, the time for the observations was determined by the interval between the activities of replacing the concentrate to the animals and those of cleaning the facilities. Thus, behaviour of all lambs was recorded just after bottle feeding (~ 8:30 a.m.) till four hours later.”

3/ line 107-111- Was excretion behavior analyzed? if so, please add it in the method

Answer: No, the excretion behavior was not studied in the trial. We agree with you that it would have been an interesting behavior to monitor.

4/ line 125-126 please specify average daily gain, average feed intake and feed conversion ratio in particular groups. Were leftovers of uneaten feed controlled?

Answer: This query has been clarified at the beginning of the Results & Discussion section. Orts were daily controlled.

5/ line 135-136 in the description of table 1, please explain whether the time spent on individual activities in a given week is given as a daily or weekly result?

Answer: We have change the sentence following your indications. In the new version is as follows: “Table 1 shows the effects of group, time (week) on fattening and their interaction on frequency of drinking behaviour and average duration of each episode elicited in eating and resting behaviours.”

Reviewer 3 Report

Article 2081636

The aim of the manuscript presented  by  Ana González-Martínez et al. was to study the effects of extending milk replacer feeding along with solid feed during the fattening period on the behaviour and welfare of lambs.

Comments:

The title need correction.

The terms weaned and unweaned suggest that one group stayed with their mothers and the other did not. It is better to write a group receiving milk or supplemented with milk and without a milk supplementation.

Line 94-96 Did all lambs reach a body weight of 25 kg in the same time?

Please specify  average daily gain, average feed intake, and feed conversion ratio for each pen.

Why were observations limited to 4 hours after milk administration?

Tables need editorial correction.

Author Response

RESPONSE

We would like to thank for your work and valuable comments that have substantially helped us to improve the former manuscript quality. All your comments have been considered and corrections have been made according to them. To make your work easier, we have highlighted the changes you and rest of reviewers requested by using track changes function.

We hope that you like the new version of the manuscript.

Thank you very much for your interest.

Kind Regards,

The authors

1/ The title need correction.

Answer: The title has been changed. It is as follows: “Effects of extending milk replacer feeding during the fattening period on the behaviour and welfare of lambs. A preliminary  study.”

2/ The terms weaned and unweaned suggest that one group stayed with their mothers and the other did not. It is better to write a group receiving milk or supplemented with milk and without a milk supplementation.

Answer: We have agreed with your indications, thus we have clarified the text in the following sense: “the group that was fed daily a bottle of milk

3/ Line 94-96 Did all lambs reach a body weight of 25 kg in the same time?

Answer: No. Final experimental body weight was fixed at 25 kg as average. This point has been in the new version.

4/ Please specify  average daily gain, average feed intake, and feed conversion ratio for each pen.

Answer: This query has been clarified at the beginning of the Results & Discussion section.

5/ Why were observations limited to 4 hours after milk administration?

Answer: This was motivated to prevent other stimuli interfering in the experience. In this sense, we have added the following phrase in the text: “In order to prevent other stimuli interfering in the test, the time for the observations was determined by the interval between the activities of replacing the concentrate to the animals and those of cleaning the facilities. Thus, behaviour of all lambs was recorded just after bottle feeding (~ 8:30 a.m.) till four hours later.”

6/ Tables need editorial correction.

Answer: We have corrected the detected errors.

Reviewer 4 Report

Authors study the effects of prolonged artificial milk feeding during fattening with a high concentrate diet on the behaviour and welfare of lambs.

The title indicates the aim of the manuscript and the abstract is well written. It clearly indicates the work objective, methodology and result of the study.

The introduction is also well written.

The objectives of the study are of interest and are in line with the scope of the journal.

The manuscript is well organized. The methodology is well articulated and the description is well made.

The conclusions are consistent with the evidence and arguments presented.

The reference is appropriate.

In my opinion, the manuscript could be accepted for publication.

Author Response

RESPONSE

We would like to thank for your work and valuable comments that have substantially helped us to improve the former manuscript quality. All your comments have been considered and corrections have been made according to them. To make your work easier, we have highlighted the changes you and rest of reviewers requested by using track changes function.

We hope that you like the new version of the manuscript.

Thank you very much for your interest.

Kind Regards,

The authors

Round 2

Reviewer 1 Report

All comments have been properly addressed and the article can be accepted in present form